# Distributed Embedded System for Multiparametric Assessment of Infrastructure Durability Using Electrochemical Techniques

**DOI:** 10.3390/s24185882

**Published:** 2024-09-10

**Authors:** Javier Monreal-Trigo, José Enrique Ramón, Román Bataller, Miguel Alcañiz, Juan Soto, José Manuel Gandía-Romero

**Affiliations:** 1Instituto Interuniversitario de Investigación de Reconocimiento Molecular y Desarrollo Tecnológico (IDM), Universitat Politècnica de València, Camino de Vera s/n, 46022 Valencia, Spain; jmonreal@upv.es (J.M.-T.); mialcan@upvnet.upv.es (M.A.); 2Departamento de Ingeniería Electrónica, Universitat Politècnica de València, Camino de Vera s/n, 46022 Valencia, Spain; robapra@alumni.upv.es (R.B.); jsotoca@upv.es (J.S.); joganro@csa.upv.es (J.M.G.-R.); 3CSIC-Instituto de Ciencias de la Construcción Eduardo Torroja (IETCC), 28033 Madrid, Spain; 4Chatu Tech S.L., 08223 Terrassa, Spain; 5Departamento de Química, Universitat Politècnica de València, Camino de Vera s/n, 46022 Valencia, Spain; 6Departamento de Construcciones Arquitectónicas, Universitat Politècnica de València, Camino de Vera s/n, 46022 Valencia, Spain

**Keywords:** durability, reinforced concrete, remote corrosion monitoring, sensor network, steel corrosion, structural health monitoring

## Abstract

We present an autonomous system that remotely monitors the state of reinforced concrete structures. This system performs real-time follow-up of the corrosion rate of rebars (i_CORR_), along with other relevant parameters such as temperature, corrosion potential (E_CORR_), and electrical resistance of concrete (R_E_), at many of a structure’s control points by using embedded sensors. i_CORR_ is obtained by applying a novel low-stress electrochemical polarization technique to corrosion sensors. The custom electronic system manages the sensor network, consisting of a measurement board per control point connected to a central single-board computer in charge of processing measurement data and uploading results to a server via 4G connection. In this work, we report the results obtained after implementing the sensor system into a reinforced concrete wall, where two well-differentiated representative areas were monitored. The obtained corrosion parameters showed consistent values. Similar conclusions are obtained with E_CORR_ recorded in rebars. With the i_CORR_ follow-up, the corrosion penetration damage diagram is built. This diagram is particularly useful for identifying critical events during the corrosion propagation period and to be able to estimate structures’ service life. Hence, the system is presented as a useful tool for the structural maintenance and service life predictions of new structures.

## 1. Introduction

Corrosion significantly impacts the durability of reinforced concrete structures (RCS) [1]. However, adherence to corrosion prevention standards is often compromised due to inaccurate exposure assessments during design or construction errors [2]. Spanish EHE 08 [3] and similar standards necessitate effective maintenance strategies to proactively address potential issues and maintain structural longevity. Timely pathology detection enables informed decision-making and corrective measures to safeguard durability and safety.

Unfortunately, inspections sometimes do not begin until the first signs of deterioration appear, which may mean that reinforcement corrosion is already advanced [4]. Portable systems that measure either the corrosion potential (E_CORR_) or corrosion rate of reinforcement offer valuable solutions [5]. While E_CORR_ provides qualitative insights, it aids in identifying potential corrosion risk zones. Table 1 outlines criteria for correlating probable corrosion risk and E_CORR_ using a calomel reference electrode (R_E_) in line with ASTM C876 [6]. The corrosion rate is the quantitative parameter that refers to loss of rebar thickness over time, typically in μm/year or current density terms (i_CORR_, as µA/cm^2^), representing the faradaic current per unit reinforcement area. Corrosion level criteria per UNE 112072:2011 [7] and ASTM STP 1065 [8], or design guideline RILEM TC 154 [9], are presented in Table 2.

Portable systems for E_CORR_ and/or i_CORR_ measurements require making electrical contact with reinforcements, limiting their implementation to accessible areas. Key devices include Gecor by Geocisa [10,11] and GalvaPulse by Germann Instruments [12]. Innovative options like Giatec’s iCOR and Andrade et al.’s proposals [13,14] have emerged. These tools often rely on guard ring systems to isolate the rebar area for testing, which might be less reliable in wet conditions [15]. Despite advancements in modulated guard-ring systems, their use still demands specialized personnel for on-site measurements, limiting their capacity for comprehensive structural monitoring [16,17].

The quest for continuous corrosion rate monitoring in RCS has spurred the development of embedded sensor systems, valued for their non-destructive nature. Among these are galvanic sensor systems, assessing corrosion risk by gauging galvanic current between paired anode-cathode electrodes [18]. Increased galvanic current signifies an altered electrochemical environment, favouring steel corrosion, often triggered by factors like elevated pore saturation and diffusion of agents like CO_2_ or chloride. Germann Instruments’ CorroWatch [19,20] is a notable commercial system, employing four anodes at varying depths to track the advancement of triggering agents through concrete covers.

Corrosion rate determination in embedded sensors entails electrochemical polarization [21]. Such sensors comprise a carbon steel working electrode (WE) mirroring the reinforcement, a stainless-steel counter-electrode (CE), and an RE, often of the MMO type (mixed metal oxide). Employing low polarizations (10–20 mV) yields polarization resistance (R_P_), convertible into corrosion current density, i_CORR_, via the linear polarization resistance (LPR) method [22]. Practical for onsite use due to its simplicity, this approach’s accuracy is acceptable [23]. However, long-term stability of the RE remains an unresolved concern [24]. Noteworthy among commercial instruments featuring these sensors are the Protector Camur-II [25] and ElectraWatch’s Embedded Corrosion Instrument (ECI) [26], both multiparameter, combining corrosion sensors with measurements like potential, resistivity, temperature, humidity, or ion presence. Rohrback Cosasco Systems’ CORRATER [27] stands out for not employing an RE, enhancing its long-term reliability. However, susceptibility to macrocell corrosion between distinct electrochemical potential areas could compromise reliability. The sensor’s WE, electrically isolated from reinforcement to confine corrosion tests, might not represent real reinforcement status due to its exclusion from such macrocells [28].

Recent decades have seen technological strides affecting corrosion sensor development. Notably, fibre optic sensors utilize reflected light intensity to gauge corrosion product formation on steel surfaces [29]. Despite their adaptability and miniaturization potential, these sensors’ long-term reliability remains unproven. Similarly, inductively coupled magnetic field-based sensors estimate reinforcement corrosion using RLC circuit resonance frequency, linked to corrosion in an enclosed steel wire [30]. Wireless measurement characterizes these sensors, enhancing installation convenience. Yet their accuracy is limited, allowing only differentiation between passive and active rebar states, impeding early corrosion detection reliability.

Recent studies have proposed enhancements to these sensor systems through wireless technology integration [31,32,33,34,35,36]. Undeniably, this represents a significant stride towards remote and real-time RCS monitoring. However, the implementation of advanced systems of this kind is still under development, with cost accessibility being the current primary challenge.

This work introduces a patented system [37] designed for remote, real-time assessment of RCS corrosion conditions, including buried and submerged areas. At each control point (CP), the system monitors reinforcement corrosion, along with key parameters like temperature and concrete electrical resistance. These measurements rely on an innovative non-destructive electrochemical polarization technique developed earlier [38]. Controlled through a central processing unit, the sensors at each CP are managed by a measurement unit with Internet connectivity, enabling remote access to monitoring data. The electronic design of the measurement unit allows for versatile exploration of various electrochemical techniques using the same hardware. This work showcases significant outcomes from the system’s validation and real-world deployment, underscoring its potential as a valuable tool for predicting structural maintenance and service life of new RCS.

## 2. Materials and Methods

### 2.1. System Overview

The presented system enables autonomous and real-time corrosion monitoring of reinforcement at multiple CPs. Embedded sensors, located at each CP, provide corrosion parameters and are connected to the measurement board using shielded paired cables. Measurement boards are situated outside the concrete structure, organized into protective boxes based on proximity. Communication between measurement boards and the central unit occurs through an RS-485 bus with a 2-wire connection. Both measurement boards and the central unit are powered by a 12 V photovoltaic system. The central unit employs a Raspberry Pi 3B (Cambridge, Cambridgeshire, UK) single-board computer with a 4G modem for Internet connection to a server. Measurement data are stored and processed on the server, accessible via a web application. The block diagram of the complete remote corrosion monitoring system is depicted in Figure 1.

### 2.2. Measurement Board

The measurement board gathers data from sensors and transmits it to the central unit. Its internal structure (Figure 2) includes an isolated buck DC–DC converter for voltage, generating a 5 V supply from the central 12 V line, further split into digital and analogue 3.3 V supplies. Communication with the central unit is achieved through an isolated RS-485 transceiver, ensuring protection against voltage surges and malfunctions while preventing direct electrical connection. 

The 32-bit ARM microcontroller manages board operations, communicating via embedded UART linked to the RS-485 transceiver. A custom protocol enables central unit control. On receiving the start measurement command, the microcontroller configures the multiplexing block and generates the needed voltage via a digital-to-analogue converter (DAC) or captures sensor signals through an analogue-to-digital converter (ADC), using 3 V voltage reference. Creating a 1.5 V virtual ground supports bipolar operation sans a negative voltage source. After sequence completion, the microcontroller awaits the central unit’s data request, then transmits the collected measurements data.

Systematic errors (offset) in the ADC and DAC are addressed through a two-step process on power-up: First, the analogue-to-digital converter (ADC) of the custom-made board is calibrated with respect to the virtual ground, which is derived from a high-precision reference voltage provided by a commercial integrated circuit. Second, the digital-to-analogue converter (DAC), responsible for signal generation in the corrosion rate sensor, is calibrated using the ADC

The system incorporates a multiplexing block with electromechanical signal relays, enabling flexible configuration of corrosion sensor measurements. Initially, the corrosion sensor’s two WEs, WE1 and WE2, are connected to the reinforcement to act as the counter-electrode (CE). During the measurement sequence, the multiplexing block establishes the appropriate signal path for each measurement, and upon sequence completion, WE1 and WE2 reconnect to the CE (reinforcement). Analogue subcircuits on the board condition and filter sensor signals before they are fed into the ADC.

### 2.3. Sensors

At each CP of the structure, a set of embedded sensors is connected to the corresponding measurement board to provide the key corrosion parameters. These sensors consist of a temperature probe, seven potentiometric sensors, and one corrosion rate sensor, all of which are assembled in a single array to be embedded close to rebars. Further details of each sensor are presented in the subsections below.

#### 2.3.1. Temperature Probe

The temperature probe consists of a K-type thermocouple combined to an integrated temperature sensor for cold junction compensation and an instrumentation amplifier. This enables the measurement of the concrete temperature within a range between −10 and 100 °C with an error of ±0.5 °C.

#### 2.3.2. Potentiometric Sensors

Seven potentiometric measurements are included in the system. A MnO_2_ electrode is used as the RE for the potentiometry measurements. It consists of three compartments, namely a porous hydrated cement paste as a bottom layer, conductive alkaline slurry as a middle layer, and MnO_2_ as a top layer. This is the ERE20 electrode commercialised by Force Technology (Brøndby, Denmark).

Of the seven potentiometric channels, three are used to measure the corrosion potential (E_CORR_): one to measure the reinforcement’s E_CORR_ and two to measure the E_CORR_ of the two WEs (WE1 and WE2). The other four potentiometric channels could be used for other potentiometric sensors (i.e., chloride and pH sensors [39,40]).

For each channel, the potentiometry circuit consists of a high impedance voltage follower to avoid drawing current from the sensor. This circuit is connected to a second-order Sallen–Key low-pass filter (LPF) to reduce noise before inputting the signal to the ADC.

#### 2.3.3. Corrosion Rate Sensor

The corrosion rate sensor (see diagram at Figure 3) operates by means of the electrochemical polarisation of a WE made of the same material as the reinforcement of the structure to be monitored. In standard structures, the WE consists of a piece of carbon steel corrugated bar, whose ends are encapsulated in an epoxy resin-filled PVC piece to delimit the working area and to protect the electrical connection to the copper wire installed on one of the ends. For redundancy, two WEs (WE1 and WE2) are included in the corrosion sensor of each CP.

Measurements on the corrosion rate sensor are taken in a two-electrode configuration, eliminating the need for a RE. Despite the performance of MMO-type electrodes currently being promising, suppressing the RE increases robustness, facilitates its installation, and favours its durability.

### 2.4. Operational Principle of the Corrosion Rate Sensor

The corrosion rate sensor has a central importance in the proposed corrosion monitoring system. Figure 4 shows the two states of the corrosion rate sensor: idle state (I) and measurement state (II). In the idle state, no measurement is taken and the WE remains connected to a nearby rebar. In this way, if macrocell corrosion processes occur in the structure, they will also affect the WE, and a macrocell current (i_MACRO_) proportional to that generated in the adjacent reinforcement region will flow through it.

When measuring the corrosion rate, the sensor passes to state II. In this case, the WE is temporarily disconnected from the reinforcement, which now acts as the CE. In this way, the two-electrode cell needed to take the electrochemical polarisation measurement is generated. The employed technique is an innovative potential step voltammetry (PSV) approach, which was described and validated in previous works [38] and whose main processes are found in the scheme shown in Figure 5. It bases the i_CORR_ measurement on the Tafel intersection method but offers the advantage of Tafel lines being obtained much more quickly and, most importantly, poses no risk of producing irreversible rebar damage.

Each Tafel line is defined by three points (Figure 5c), which result from applying the potentiostatic pulse sequence shown in Figure 5a. It is a symmetrical sequence that alternates anodic (+ΔE) and cathodic steps (−ΔE) with the inclusion of zero-amplitude steps between them to return the steel–concrete system to its original open circuit potential (OCP). The applied potentials ±ΔE1, ±ΔE2, and ±ΔE3 correspond to ±70, ±105, and ±140 mV. The OCP corresponds to the WE potential measured versus the CE before polarisation. Each point on the Tafel lines is obtained by modelling the system’s transitory response to the respective ∆E pulse with a specific equivalent circuit (Figure 5b). Circuit components are calculated by the least squares fitting of the current–time response. This provides the faradaic current (I_F_) that passes through the system (the point ordinate) and the overpotential (η) applied to the respective step (the point abscissa). Once both Tafel lines (anodic and cathodic) are built, i_CORR_ is obtained from their intersection (Figure 5c). The electrical resistance of concrete (R_E_) and double-layer capacity (C_DL_) are relevant corrosion parameters and are also calculated from the circuit components and considering the working electrode area (S_WE_), as shown in Figure 5b. In summary, the corrosion sensor provides several relevant corrosion parameters, i.e., i_CORR_, C_DL_, and R_E_, in a single measurement.

The PSV technique used to assess corrosion parameters is a two-step measurement. Firstly, the OCP of the WE vs. the CE must be obtained by using the potentiometry circuit described in Section 2.3. Secondly, after being adjusted with the corresponding obtained OCP value, the pulse sequence of Figure 5a is applied and the current generated on the WE is measured. To this end, the microcontroller generates the voltage corresponding to potential pulses by means of the embedded DAC and applies it to the CE. An operational amplifier in the voltage follower configuration is used to amplify the DAC output current. In parallel, a transimpedance amplifier is employed to convert the current generated at the WE (when pulses are applied) into a voltage. The current-to-voltage converter has two scales, ±1 mA and ±10 μA, which can be selected by an analogue switch. The obtained voltage is then passed through the LPF and inputted to the ADC. The resulting signal is finally sampled by the microcontroller. While taking the measurement, the microcontroller establishes the connection between the WE and the CE to the corresponding analogue subcircuits by means of the relays of the multiplexing block. The data is sent to the central unit at the end of the measurement sequence where they are converted into current values and used to calculate the corrosion parameters by the described methodology.

### 2.5. Analysis Tool

The measurement board at each CP provides a data file for every measurement, which contains the response recorded for the different sensors described in Section 2.3. Data are processed by an R algorithm, which is specifically developed to obtain the parameters sought for each sensor. This algorithm is automatically run daily from the system’s central unit (Figure 1). The flow chart in Figure 6 shows the processes included in the algorithm to structure and analyse the information in the data file obtained from the measurement board. 

Temperature, potentiometric, and voltammetric data is analysed. Regarding the last, 1300 records from each channel correspond to the intensity response vs. time (I-t) obtained when applying the potentiostatic pulse sequence seen in Figure 5a in the corresponding WE of the corrosion sensor (WE1 and WE2). This is when a current value is recorded every 0.5 s for the 650 s that measuring lasts. Signal I-t is processed to determine the R_E_, C_DL_, and i_CORR_ values following the steps described in Section 2.3.3. The first step is that represented in Figure 5b; that is, the least squares fitting of the current-time response of all six pulses (+ΔE1, +ΔE2, +ΔE3, −ΔE1, −ΔE2, and −ΔE3) so that, with the fitted value of the equivalent circuit components, the overpotential (η), faradaic current (I_F_), R_E_, and C_DL_ are obtained.

At this point, checks are made to see if the fit for each pulse is valid. To do so, whether the value of the circuit components obtained from the fit is not abnormal is verified (negative values or with higher orders of magnitude than 106) and if the resulting R^2^ coefficient is over 0.8. If fitting is successful in at least two anodic pulses +ΔE) and two cathodic pulses (−ΔE), the two Tafel lines can be built and i_CORR_ can be obtained as shown in Figure 5c, as long as (i) the slope of each straight line is coherent, positive on the anodic line, and negative on the cathodic line; (ii) the coordinate of the intersection (η) equals 0.00 ± 0.01 V. If condition (i) is not met on one of the lines, i_CORR_ is obtained from the intersection of the other line in η = 0. The same procedure applies if condition (ii) is not met, but the anodic line is always used in this case. Should at least two I_F_-η values not exist for some lines, attempts are made to build the other line, whose intersection is η = 0. If condition (ii) is met, it provides i_CORR_. In the other assumptions, the measurement is considered invalid and there is no result for i_CORR_. Both R_E_ and C_DL_ are obtained from the average of the values found in valid pulse fits. The finally obtained parameters are uploaded to the server’s database where they can be remotely consulted in real time.

### 2.6. Reinforced Concrete Structure

The number of Control Points (CPs) and the distance between them depend on the specific characteristics of the structure being monitored. It is common practice to create representative batches based on structural typology, exposure conditions, and damage level, following a strategy similar to that described in reference manuals, such as the CONTECVET manual [41]. For example, in the case of a bridge over the sea, the batches might include the following:Submerged portions of the pylons;Portions of the pylons in the tidal range zone;Portions of the pylons above the tidal range zone;The bridge deck.

Typically, not all elements within each batch are monitored; instead, a representative sample is selected. The number of elements to instrument usually depends on the type of structure (e.g., whether it is critical infrastructure or a private project) and the desired level of statistical control. This decision is often made in consensus with the structure’s owner, who sets a general criterion based on available resources and specific interests.

In the case of the prototype structure presented in this article, two working electrodes (WEs) are installed at each CP, effectively doubling the corrosion measurement results at each CP. This approach ensures some degree of reproducibility, and in the worst-case scenario, if one WE fails, monitoring can continue with the other. Regarding the distance between WEs at a CP, there is no fixed rule, although it is preferable that they are not too far apart to ensure comparable results and that their position relative to the reference electrode is as close as possible. However, the distance should be sufficient to allow the concrete to flow during construction. In practice, this distance depends on the specifics of the element being monitored, generally influenced by the number and arrangement of the reinforcement, which dictates the available space for installing each WE.

For validation, a concrete wall was built with embedded sensors in CPs (Figure 7). Measuring 3 m in length, 0.15 m in thickness, and 1.3 m in height, it stood on a spread footing (3.4 m × 0.55 m × 0.3 m). Reinforcement comprised B500SD carbon steel rebars (Ø 10 mm) forming a 15 cm spaced mesh. Concreting occurred in two stages, up to an intermediate height, using the specified dosage shown in Table 3.

In the second phase, the upper wall level was filled with chloride-contaminated concrete. This aimed to create distinct electrochemical environments for sensor response evaluation: (i) an aggressive zone (chloride-contaminated upper area) encouraging corrosion onset; (ii) a non-aggressive zone (chloride-free lower area) with no initial corrosion risk. To make chloride-contaminated concrete, Table 3 dosage was applied, with the addition of 35 g/L NaCl to the mixing water.

Three CPs were set up on the wall (Figure 7d): CP1 at the top (chloride-contaminated), and CP2 and CP3 at the bottom (non-aggressive). To enable simultaneous and automated sensor monitoring at these CPs, the electronic system described in Section 2.1 and Section 2.2. Here, all measurement boards were placed within an airtight box adjacent to the wall (Figure 7f), alongside the power and control (Figure 7g). The measurement system was programmed for daily readings at 12:00 h.

## 3. Results

### 3.1. Electrical Validation of the Measurement Platform

The accuracy of potentiometric value acquisition, voltage signal generation, and current measurement is validated against a standard reference instrument (Keithley 2000 [42]). The potentiometric sensor channels are tested for values from 400 mV to 100 mV and from −400 mV to −100 mV in increments of 100 mV. For evaluating DAC output voltage generation and current measurement accuracy, output voltages of ±70, ±105, and ±140 mV are applied. The current measurement is assessed in two scales: ±1 mA using a 280 Ω resistor and ±10 μA using a 28 kΩ resistor, applying the same voltage steps. In Table 4, the average values and standard deviation for each parameter’s relative error are shown. The results confirm high accuracy, with deviations well within acceptable limits.

### 3.2. Current Response Signal for Corrosion Rate Measurements

The temporal evolution of the current responses collected by the system are shown in Figure 8**.** The corrosion rate measurements at CP1, which presents higher concentrations of chlorides (active state, Figure 8a), are compared to the CP2 ones (without introduced aggressive ions, i.e., passive state, Figure 8b). The low noise level and good match between measured and fitted curves is represented by the normalised root mean square error (NRMSE) of the measured current response in relation to the fitting curve. Table 5 shows the calculated NRMSE values.

On the one hand, a low noise level was achieved by using a second-order Sallen–Kay low-pass filter with a cut-off frequency of 50 Hz and the ADC oversampling and averaging. On the other hand, the good correspondence between the current response signal and the fitting curves validates the equivalent circuit proposed in [38] for assessing I_F_, R_E_, and C_DL_.

### 3.3. Analysis of the Monitored Durability Parameters

This section examines the evolution of monitored parameters over 280 days at CPs (CP1, CP2, and CP3) on the wall: temperature (T), concrete’s electrical resistance (R_E_), corrosion potential (E_CORR_), and corrosion current density (i_CORR_). Each CP recorded both E_CORR_ of the WEs and the rebar’s E_CORR_. The temperature trends (Figure 9) were similar at all CPs, with an average coefficient of variation (CV) of 17.4%. The median average percentage error (MAPE) between CP values and Valencia-Airport Weather Station records [43] was 39.5%, 54.9%, and 47.5% for CP1, CP2, and CP3, respectively. Considering the station’s distance (10 km) and altitude difference (90 m), these differences are acceptable.

When analysing the corrosion parameters, Figure 10, Figure 11 and Figure 12 indicate that observed oscillations were linked to temperature and humidity variations, both influential in corrosion kinetics. Humidity had a notable impact, with higher humidity leading to lower R_E_, more negative E_CORR_, and increased i_CORR_, signifying accelerated corrosion kinetics. 

In the different parameters, the evolution of the corrosion system’s two WEs (WE1 and WE2) at each CP was similar (Figure 10, Figure 11 and Figure 12). The average values for the monitored period, found in Table 6, were also similar. 

## 4. Discussion

CP parameter analysis reveals values coherent with respective exposure conditions. In the chloride zone (CP1), the average R_E_ was 3114.7 Ω, while the chloride-free zones (CP2 and CP3) had higher R_E_ at 6440.5 Ω and 5708.1 Ω, respectively. E_CORR_ and i_CORR_ differentiation between zones is clear. CP1, the chloride-contaminated zone, showed high corrosion risk (E_CORR_ between −370 and −627 mV) and mainly moderate i_CORR_ (0.5–1.0 µA/cm^2^), varying with humidity. CP2 and CP3, chloride-free zones, exhibited low-risk E_CORR_ (>−124 mV), except during heavy rainfall (≈180–280 days), leading to higher i_CORR_ (Figure 12), but rebars remained passive (i_CORR_: 0.015–0.052 µA/cm^2^), on average 19-fold lower than CP1. Figure 13 compares the average E_CORR_ of sensor’s WEs (WE1, WE2) to direct reinforcement measurements, showing similar trends over time for both sensors and reinforcement.

i_CORR_ determined corrosion state quantitatively, while both i_CORR_ and E_CORR_ remained coherent, supported by Figure 10 and Figure 11 and Table 6. In the chloride-free areas (CP2, CP3) with rising humidity, corrosion level was negligible (i_CORR_)-low (E_CORR_). In the chloride-rich zone (CP1) with generally dry conditions, corrosion level was moderate (i_CORR_)-high (E_CORR_). i_CORR_ stood out among all parameters, essential in concrete standards like EN 1992 Eurocode 2 [44] for service life prediction. It estimates time before structure repair due to corrosion damage, reflecting the extent of the propagation period (t_P_) in Tuutti’s model [45]. Similarly, corrosion penetration damage (P_X_) at certain times during t_P_ was deduced from i_CORR_ using Equation (1) [46].
P_X_ (µm) = 11.6·i_CORR_·t_p_(1)

Figure 14 displays P_X_ values over time for each CP, derived from corresponding i_CORR_ values in Figure 12. For CP2 and CP3, where rebars remained passive, the propagation period had not begun, accounting for the gradual slopes in P_X_-t graph. CP1 exhibited a more pronounced slope, as the propagation period started due to chlorides in the concrete mix, maintaining active rebars throughout monitoring (Figure 11 and Figure 12). P_X_ at CP1 was notably below the 75 µm threshold from the literature [47], beyond which surface cracks could appear (0.3–0.4 mm). This limit’s applicability depends on factors like concrete density, rebar diameter, and cover depth. Slope changes in CP1′s P_X_ evolution correlated with corrosion kinetics changes prompted by climatic shifts, highlighted in i_CORR_ evolution analysis (Figure 12).

The representation of Figure 14 is also practical for determining rebar diameter loss (∆*ϕ*x) using Equation (2) [48], with *α* as the pitting factor (2 for uniform corrosion, 3–10 for pitting corrosion) [9].
∆*ϕ*x (µm) = *ϕ*_0_ −P_X_·α(2)

The corrosion sensor holds utmost significance among the implemented sensors, demanding precise measurements. Low NMRSE values (approximately 2% at CP1 and 0.2% at CP2) validate the minimal noise in the current–time signal and the strong correlation between the experimental response and the fitting curves for i_CORR_ determination.

The recorded parameters have clearly differentiated values in the two evaluated zones and are consistent with the environmental climate conditions (temperature and humidity), but especially with the design conditions for validation. In the chloride-contaminated zone, i_CORR_ ≈ 0.524 µA/cm^2^, E_CORR_ ≈ −0.472 V vs. SCE, and R_E_ ≈ 3115 Ω are obtained (on average), which indicates an active state. In the chloride-free zone, i_CORR_ ≈ 0.028 µA/cm^2^, E_CORR_ ≈ −0.075 V vs. SCE, and R_E_ ≈ 6074 Ω are obtained and indicate a passive state.

The deviations between the corrosion sensor’s two WEs installed at each CP, plus their deviation in relation to the reinforcement (for E_CORR_), are not important because the corrosion risk level associated with the E_CORR_ and i_CORR_ values is in keeping in each zone: a negligible-low risk in the chloride-free zone and a moderate-high risk in the chloride-contaminated zone.

## 5. Conclusions

We introduce an autonomous system for remote real-time corrosion monitoring in concrete structures across various control points (CPs). Successfully validated on a reinforced concrete wall, the system monitored three CPs representing different areas: chloride-contaminated (CP1) and chloride-free zones (CP2–CP3).

Each CP provides corrosion rate (i_CORR_), corrosion potential (E_CORR_), electrical resistance of concrete (R_E_), and temperature (T) measurements through embedded sensors. These responses are recorded using a custom measurement board and analysed with a dedicated R-based data analysis application. Managed by a central module, the sensor network operates autonomously, powered by photovoltaics and connected to a 4G modem for internet access. Data is uploaded to a server for user access. The measurement board’s hardware is designed to be capable of implementing voltammetric techniques currently under research.

Of the monitored parameters, i_CORR_ is especially interesting for being the parameter employed in models that estimate structures’ service life. The thorough follow-up offered by the system allows the effects of seasonal cycles on i_CORR_ to be observed to provide a representative value. Moreover, integrating the i_CORR_ graph gives corrosion penetration damage (P_X_), which is a parameter of acknowledged usefulness for identifying the critical moments of the propagation period, such as cracks appearing, and for determining loss of rebar diameter. This makes the presented monitoring system extremely useful, not only from the practical viewpoint to be able to evaluate the corrosion condition of the structures but also from the perspective of research and development of durability models with new materials.

The technology developed is designed to allow the incorporation of all types of potentiometric and voltammetric sensors, including the promising oxygen and chloride sensors currently under development. All the advantageous features of the system presented here have aroused some interest in the industrial sector.

## Figures and Tables

**Figure 1 sensors-24-05882-f001:**
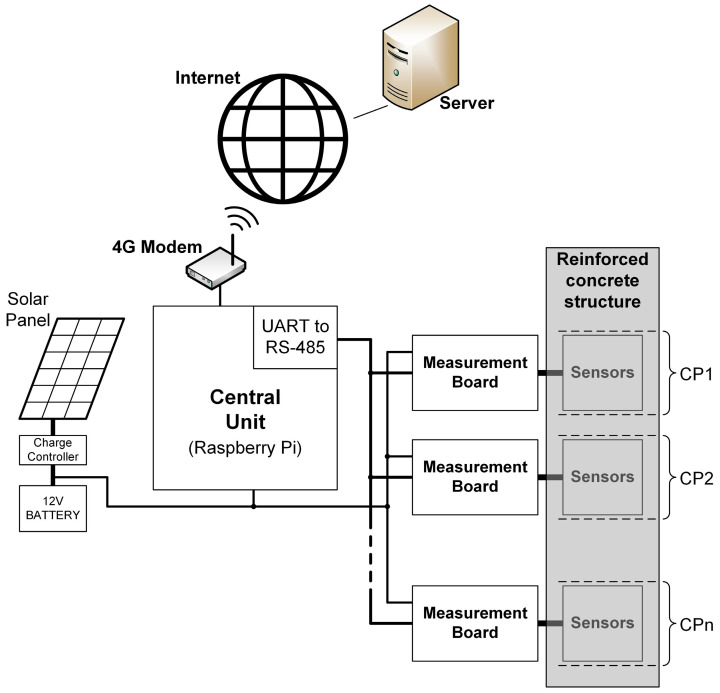
Block diagram of the remote corrosion monitoring system.

**Figure 2 sensors-24-05882-f002:**
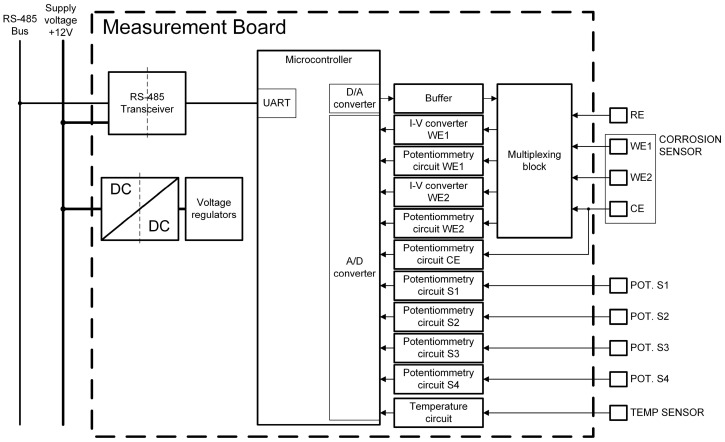
Block diagram of the measurement board.

**Figure 3 sensors-24-05882-f003:**
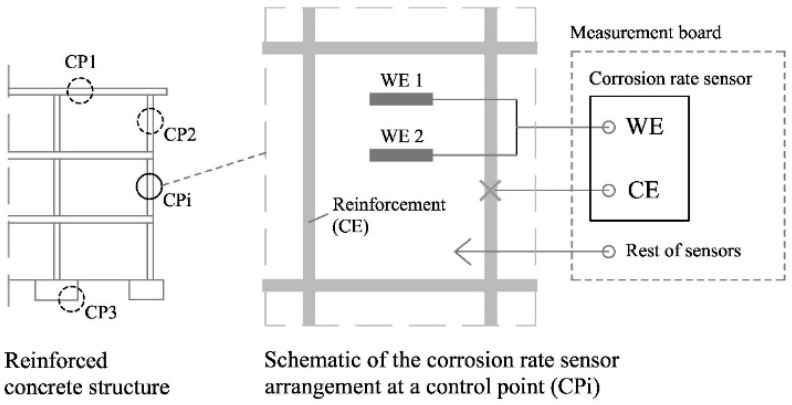
Block diagram of the corrosion rate sensor in the RCS and at one CP and its connection to the measurement board.

**Figure 4 sensors-24-05882-f004:**
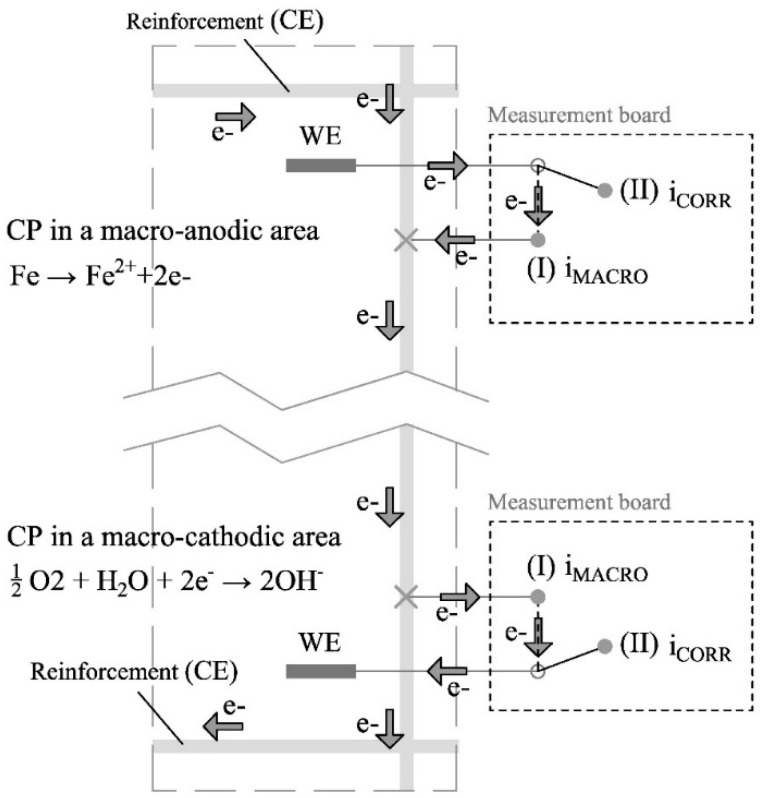
Corrosion rate sensor states.

**Figure 5 sensors-24-05882-f005:**
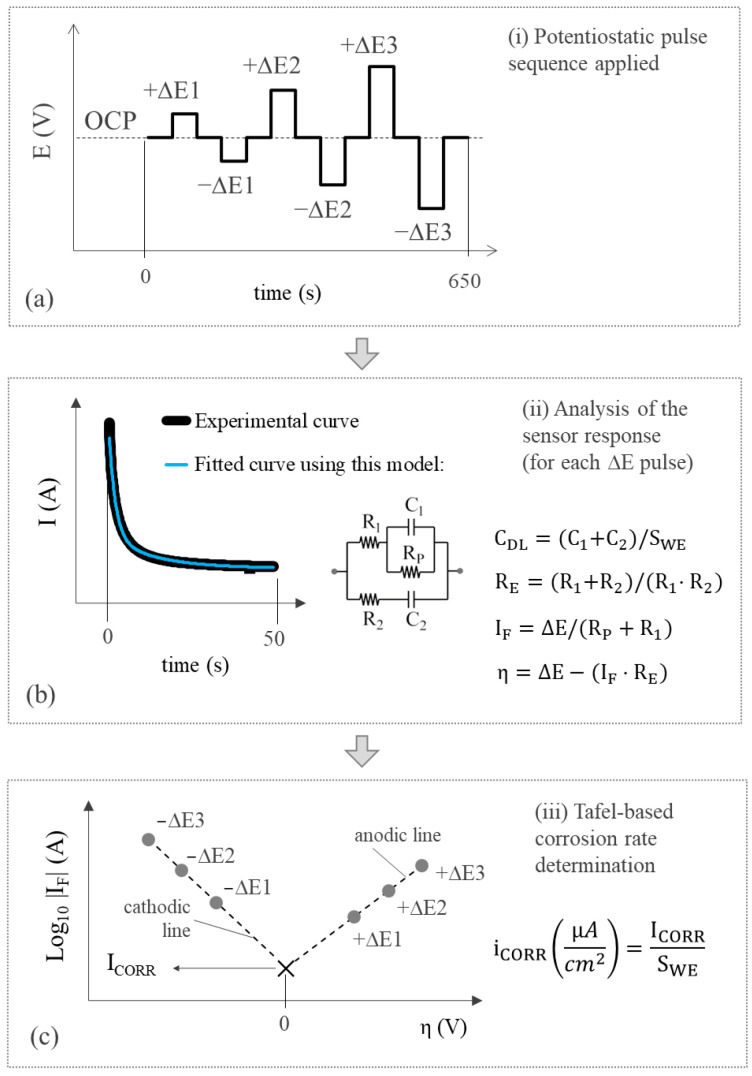
PSV-based corrosion rate measurement technique. (i) Potentiostatic pulse sequence applied, (ii) Analysis of the sensor response (for each ΔE pulse), (iii) Tafel-based corrosion rate determination.

**Figure 6 sensors-24-05882-f006:**
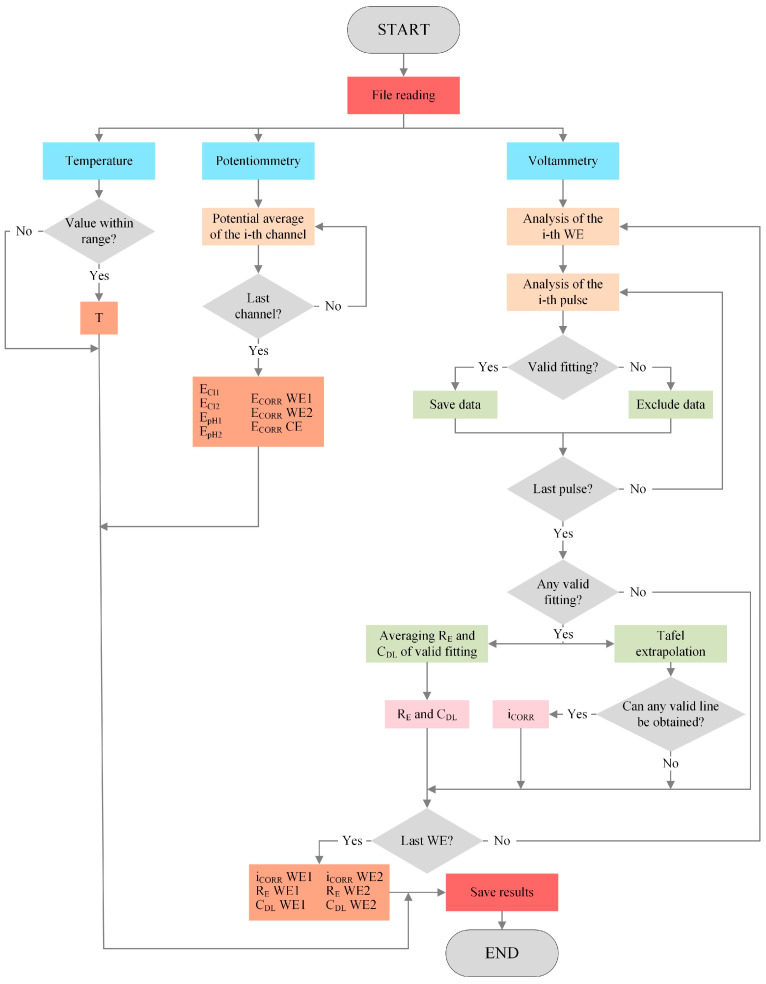
Flow chart of the algorithm for structuring and analysing the information in the data file obtained from the measurement board.

**Figure 7 sensors-24-05882-f007:**
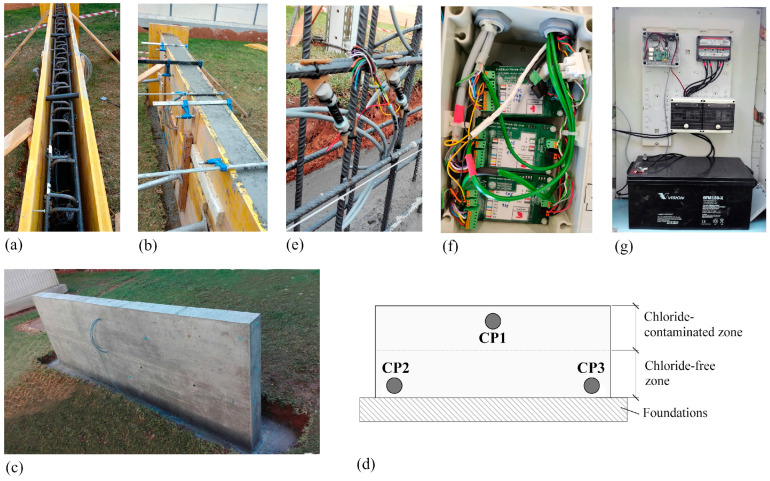
Wall used in validating the monitoring system: (**a**,**b**) execution process, (**c**) wall executed, (**d**) schematic diagram of the control points (CP) location, (**e**) corrosion sensor assembled to the wall reinforcement at one of the CPs, (**f**) measurement boards into an airtight box, and (**g**) cabinet with the central unit, along with the storage, regulation, and control devices for the photovoltaic system.

**Figure 8 sensors-24-05882-f008:**
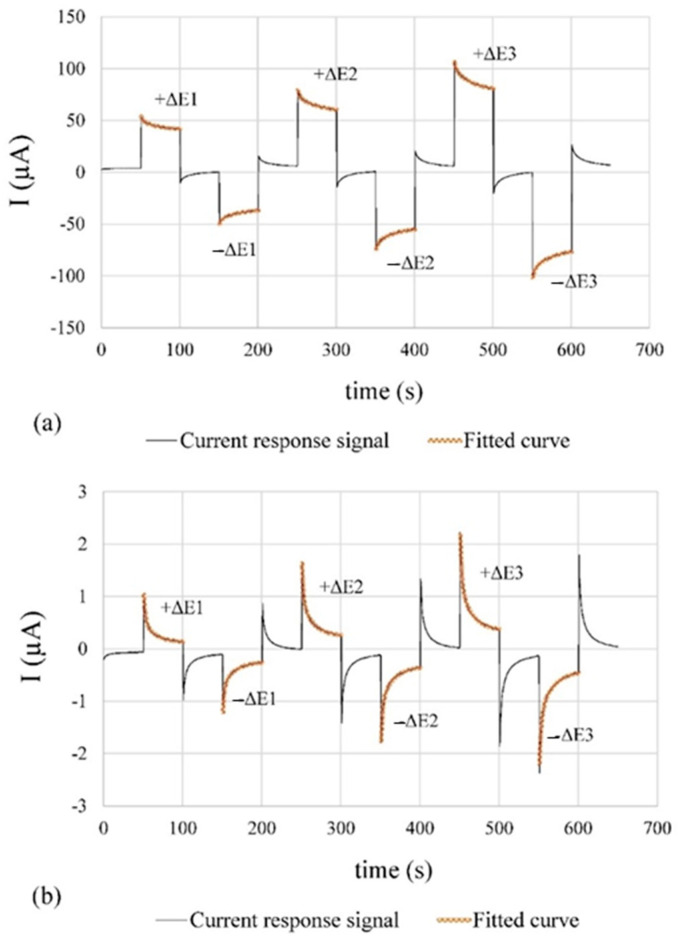
Current response signal to the sequence of the potentiostatic pulses in Figure 5a and the fitted curve for the anodic and cathodic pulses (±ΔE): (**a**) CP1, active corrosion state; (**b**) CP2, passive corrosion state. In both cases, the signal corresponds to WE1.

**Figure 9 sensors-24-05882-f009:**
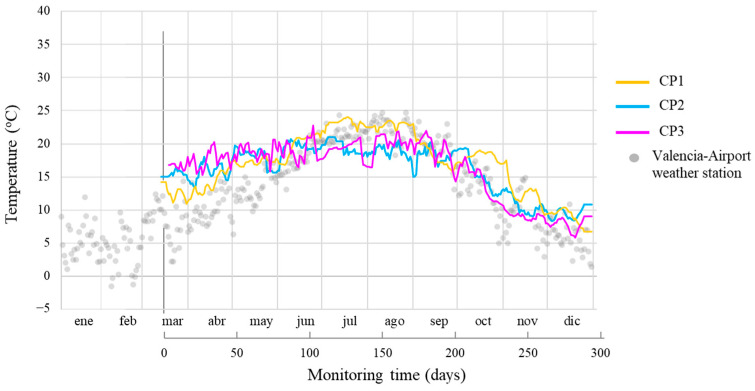
Temperature measurements of CP1, CP2, and CP3 vs. the Valencia-Airport weather station logs.

**Figure 10 sensors-24-05882-f010:**
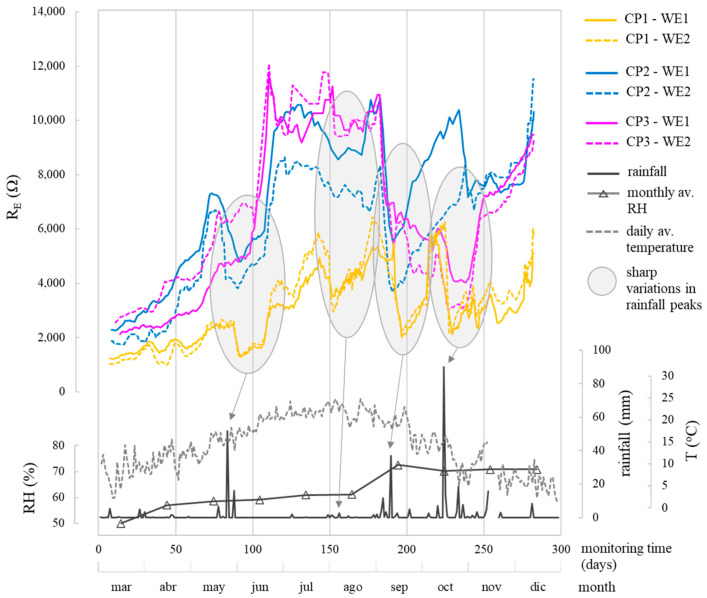
R_E_ estimation over each CP, rainfall, temperature, and relative humidity (RH) over time. Sharp variations in R_E_ are related to rainfall peaks.

**Figure 11 sensors-24-05882-f011:**
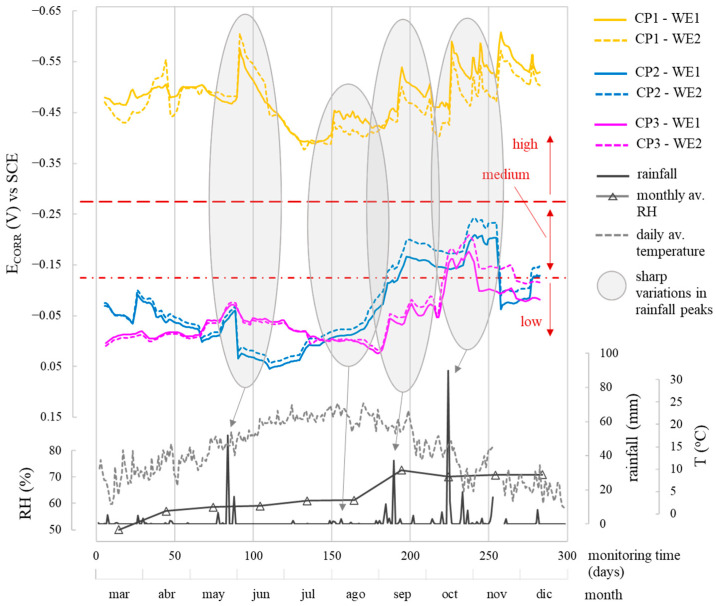
Evolution of the corrosion potential (E_CORR_) recorded by the sensor system at CP1, CP2, and CP3 by the two WEs of the installed corrosion sensor (WE1 and WE2). The profiles of rainfall, mean monthly relative humidity, and mean daily temperature recorded by the Valencia-Airport Weather Station are provided.

**Figure 12 sensors-24-05882-f012:**
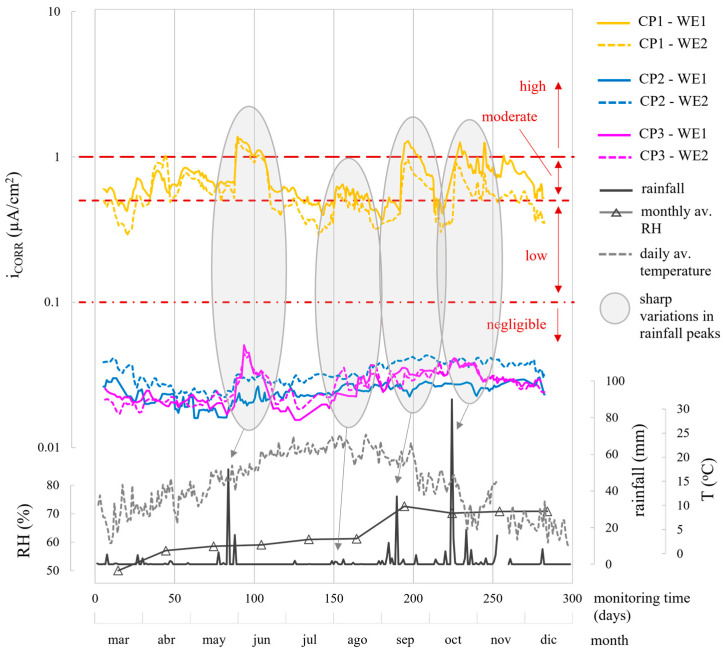
Evolution of the corrosion current density (i_CORR_) recorded by the sensor system at CP1, CP2, and CP3 by the two WEs of the installed corrosion sensor (WE1 and WE2). The profiles of rainfall, mean monthly relative humidity, and mean daily temperature recorded by the Valencia-Airport Weather Station are provided.

**Figure 13 sensors-24-05882-f013:**
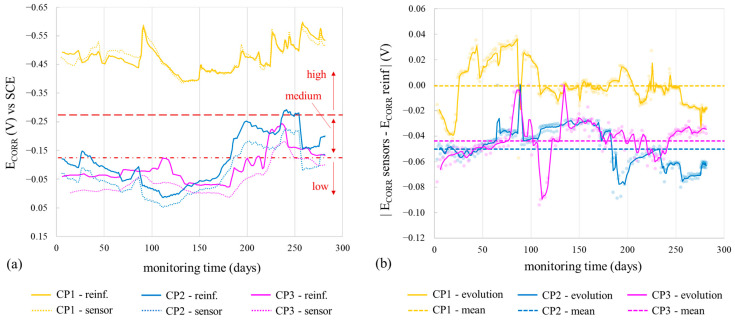
Comparison between the average corrosion potential (E_CORR_) of the WEs (sensor) and that of the reinforcement (reinf.) at CP1, CP2, and CP3: (**a**) evolution of E_CORR_ with time and (**b**) the difference between the sensor’s E_CORR_ and that of the reinforcement as an absolute value (|E_CORR_ sensor − E_CORR_ reinf|) vs. time.

**Figure 14 sensors-24-05882-f014:**
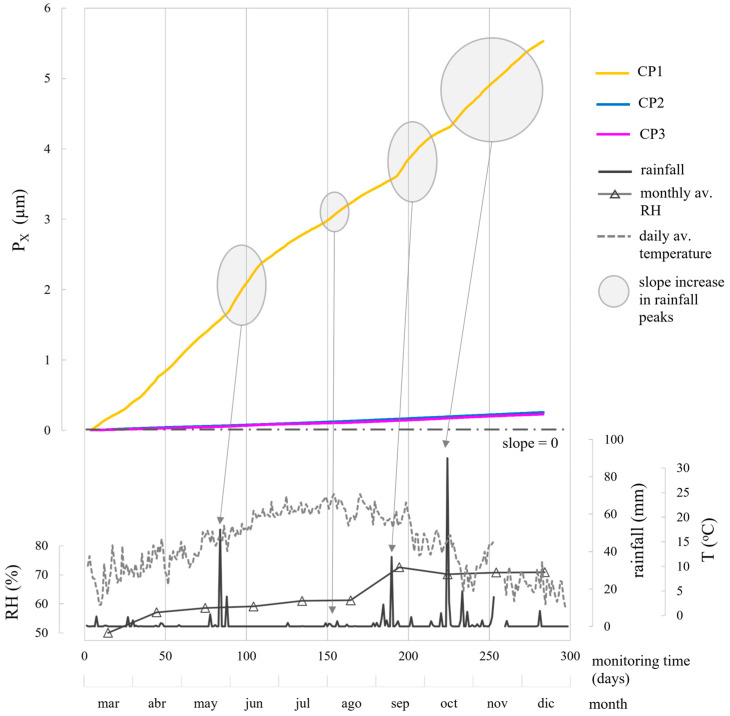
Evolution of corrosion penetration damage (P_X_) determined according to Equation (1) from the i_CORR_ values (average of WE1 and WE2) at CP1, CP2, and CP3. The profiles of rainfall, mean monthly relative humidity, and mean daily temperature recorded by the Valencia-Airport Weather Station are provided.

**Table 1 sensors-24-05882-t001:** Corrosion risk criteria based on the E_CORR_ value measured by the saturated calomel electrode (SCE) according to [6].

*E_CORR_* (V)	Corrosion Risk
>−0.124	low (<10%)
from −0.274 to −0.124	medium (≈50%)
<−0.274	high (>90%)

**Table 2 sensors-24-05882-t002:** Corrosion level criteria based on the corrosion rate and i_CORR_ value according to [7,9].

*i_CORR_* (μA/cm^2^)	Corrosion Rate (μm/Year)	Corrosion Level
<0.1	<1.16	negligible
from 0.1 to 0.5	from 1.6 to 5.8	low
from 0.5 to 1.0	from 5.8 to 11.6	moderate
>1.0	>11.6	high

**Table 3 sensors-24-05882-t003:** Concrete mix design as kg/m^3^.

Concrete Mix	Density (kg/m^3^)
Cement CEM II-/B-M(S-L) 42.5R	335
Water	218
Sand 0/2	579
Sand 0/4	579
Gravel 4/8	579

**Table 4 sensors-24-05882-t004:** Accuracy and standard deviation of the custom-board measurement platform vs. reference Keithley 2000 [42].

	Potentiometric Channels	Digital-to-Analogue Output Voltage	Current Measurement in ±1 mA Scale	Current Measurement in ±10 µA Scale
Average	0.1%	0.4%	0.3%	0.3%
Std. dev.	±0.6%	±0.7%	±0.7%	±0.6%

**Table 5 sensors-24-05882-t005:** NRMSE between the current response signal and the fitting curves obtained in CP1 (active corrosion state) and CP2 (passive corrosion state).

Potentiostatic Pulse	NRSME (%) CP1	NRSME (%) CP2
+ΔE1	0.17	2.08
−ΔE1	0.22	1.49
+ΔE2	0.18	1.64
−ΔE2	0.15	1.44
+ΔE3	0.15	1.65
−ΔE3	0.14	2.33

**Table 6 sensors-24-05882-t006:** Average values of concrete’s electrical resistance (R_E_), corrosion potential (E_CORR_), and corrosion current density (i_CORR_) for the whole monitoring period in both the corrosion sensor’s WEs (WE1 and WE2) at each CP and the general average of the CP (average of WE1 and WE2). The corrosion risk associated with the average value of each parameter is indicated according to [6,7,9].

		*R_E_* (Ω)		*E_CORR_* (V) vs. SCE			*i_CORR_* (µA/cm^2^)	
		WE1	WE2	WE1	WE2	Reinf	WE1	WE2
**Mean**	**CP1**	2932	3297	−0.482	−0.461	−0.473	0.546	0.501
	**CP2**	6933	5948	−0.062	−0.077	−0.120	0.024	0.032
	**CP3**	5859	5558	−0.060	−0.098	−0.098	0.027	0.027
	**CP1**	3115		−0.471		−0.473	0.524	
	**CP2**	644		−0.070		−0.120	0.028	
	**CP3**	5708		−0.079		−0.098	0.027	
**Corrosion risk**	**CP1**	-		High		High	Moderate	
	**CP2**	-		Low		Low	Negligible	
	**CP3**	-		Low		Low	Negligible	

## Data Availability

The original data presented in the study are openly available in FigShare at https://doi.org/10.6084/m9.figshare.26362372.

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
