# Peer review of "Distributed Embedded System for Multiparametric Assessment of Infrastructure Durability Using Electrochemical Techniques"

_sensors, 2024, doi:10.3390/s24185882_

Round 1
Reviewer 1 Report
Comments and Suggestions for Authors
Author Response
Please, find attached the responses in Response to Reviewer 1.docx

Reviewer 2 Report
Comments and Suggestions for Authors
It would be beneficial if the authors specified the values of the potential steps.
Some of the figures are very small and difficult to interpret. In particular, it is suggested to enlarge Figures 5 and 7.
Some acronyms should be defined the first time they are used. For example, CP (line 104) and SWE (Figure 5).
Comments on the Quality of English LanguageI am not a native English speaker, but it seems to me that the article is very well-written in English.
There are likely minor grammatical issues that I might not detect.
In particular, I am intrigued by the phrase in the abstract (lines 19 and 20), where it is stated that the sensor 'is stressed.' What do the authors mean by this?
Author Response
Please, find attached the responses in Response to Reviewer 2.docx

Reviewer 3 Report
Comments and Suggestions for Authors
The work presents a novel system for real-time remote monitoring of RC structures using electrochemical methods. The integration of corrosion rate monitoring with other parameters, such as temperature, humidity and electrical resistance, offers a comprehensive approach to corrosion assessment. The reviewer recommends a minor revision before acceptance. Specific issues regarding this manuscript are as follows:
1. Line 104: Please explain the abbreviation "CP" when it first appears in the manuscript.
2. How is the distance between the embedded sensors determined when applied to actual scenarios? Please provide guidelines or criteria for sensor placement in practical applications.
3. In the introduction, the authors mention that a key challenge of current monitoring systems is the cost. How does the cost of this system compare to traditional corrosion monitoring techniques? Additionally, what is the expected lifespan of the embedded sensors and electronic components in the field?
Comments on the Quality of English Language1. Line 250: There is a writing error. Please correct it.
Author Response
Please, find attached the responses in Response to Reviewer 3.docx

Round 2
Reviewer 1 Report
Comments and Suggestions for Authors
The manuscript can be accepted.